# Silicon Supplementation Alleviates Adverse Effects of Ammonium on Ssamchoo Grown in Home Cultivation System

**DOI:** 10.3390/plants11212882

**Published:** 2022-10-28

**Authors:** Kyungdeok Noh, Byoung Ryong Jeong

**Affiliations:** 1Department of Horticulture, Division of Applied Life Science (BK21 Four Program), Graduate School, Gyeongsang National University, Jinju 52828, Korea; 2Institute of Agriculture & Life Science, Gyeongsang National University, Jinju 52828, Korea; 3Research Institute of Life Science, Gyeongsang National University, Jinju 52828, Korea

**Keywords:** *Brassica* lee ssp. *namai*, home cultivation system, hydroponics, NH_4_^+^, NO_3_^−^, silicon, Si, Ssamchoo, urea

## Abstract

Ssamchoo is recently attracting attention as a household hydroponic vegetable in Korea. It has a refreshing texture and a rich content of vitamins and fiber. Ssamchoo with a wide leaf area is suitable for traditional ssam or vegetable wraps, as well as a vegetable for salads; thus, it can be used in a variety of dishes. However, Ssamchoo plants responds sensitively to the nutrient solution, and it is often difficult to secure sufficient leaf area and robust growth using a commercial nutrient solution for leafy vegetables. This study consisted of three experiments conducted to develop the nutrient solution for Ssamchoo grown in a newly developed home hydroponic cultivation system using light-emitting diodes as the sole source of light. In the first experiment, growth and development of Ssamchoo in a representative commercial nutrient solution, Peters Professional (20-20-20, The Scotts Co., Marysville, OH, USA), was compared with laboratory-prepared nutrient solutions, GNU1 and GNU2. As a result, the Ssamchoo grown in Peters Professional had a high NH_4_^+^ content in the tissue, leaf yellowing, darkened root color, and suppressed root hair development. In addition, adverse effects of ammonium such as low fresh weight and shorter shoot length were observed. In the second experiment, Peters Professional was excluded, and the ratio of NO_3_^−^ to NH_4_^+^ in the GNU1 and GNU2 nutrient solutions was set to four levels each (100:0, 83.3:16.7, 66.7:33.3, and 50:50). As a result, the fresh weights of 83.3:16.7 and 66.7:33.3 were the greatest, and the leaf color was a healthy green. However, at 100:0 and 50:50 NO_3_^−^/NH_4_^+^ ratios, the fresh weight was low, and leaf yellowing, tip burn, and leaf burn appeared. The nutrient solution with a 83.3:16.7 NO_3_^−^- to-NH_4_^+^ ratio, which gave the greatest fresh weight in the second experiment, was chosen as the control, while the solution with a 50:50 NO_3_^−^/NH_4_^+^ ratio with a lower nitrate content among the two unfavorable treatments was selected as a treatment group for the next experiment. In the third experiment, NH_4_^+^ was partially replaced with urea to make four different ratios of NO_3_^−^ to NH_4_^+^ to urea (83:17:0, 50:50:0, 50:25:25, and 50:0:50) in combination with two levels of Si (0 and 10.7 mmol·L^−1^ Si). The greatest fresh weight was obtained in the treatment in which the NO_3_^−^/NH_4_^+^/urea ratio was 50:25:25. In particular, when Si was added to the solution, there was no decrease in the number of leaves, and plants with the greatest fresh weight, chlorophyll content, and leaf area were obtained. The number of leaves and leaf area are important indicators of high productivity since the Ssamchoo is used in ssam dishes. It can be concluded that a solution with a NO_3_^−^/NH_4_^+^/urea ratio of 50:25:25 and supplemented with 10.7 mmol·L^−1^ Si is the most suitable nutrient solution for growing Ssamchoo in the home hydroponic system developed.

## 1. Introduction

The nutrient solution in a hydroponic system is one of the most important factors that determines the yield and quality of plants. This is a solution in which inorganic ions, essential elements of higher plants, are dissolved. Essential elements for plants have distinct physiological roles, and their deficiency or absence can prevent complete plant growth and life cycle. Elements essential for plants can be divided into macro- and micro-elements. Macro-elements are carbon (C), hydrogen (H), oxygen (O), nitrogen (N), phosphorus (P), potassium (K), calcium (Ca), magnesium (Mg), and sulfur (S). Micro-elements are iron (Fe), copper (Cu), zinc (Zn), manganese (Mn), molybdenum (Mo), boron (B), chlorine (Cl), and nickel (Ni). Except for carbon and oxygen, all essential elements are supplied through the growing medium. Elements such as silicon (Si), sodium (Na), vanadium (V), selenium (Se), cobalt (Co), aluminum (Al), and iodine (I) are considered beneficial elements, because they can promote growth, while counteracting the toxic effects of other elements. The most basic nutrient solution contains nitrogen, phosphorus, potassium, calcium, magnesium, and sulfur with added trace elements [1].

In hydroponics, the composition and concentration of the nutrient solution supplied to crops play a very important role. In order to obtain good quality and a high yield of cultivated plants, proper elemental composition and concentration are required.

In particular, nitrogen has a great influence on the quality and yield of cultivated plants. There are three main sources of nitrogen: nitrate (NO_3_^−^), ammonium (NH_4_^+^), and urea (CH₄N₂O) [2]. Nitrate is an essential ingredient in crops. In general, what is absorbed through the root is consumed in the leaves, and little remains unmetabolized and accumulated [3]. However, if there are more nitrate ions in the nutrient solution than necessary, they are absorbed excessively, and the remaining amount accumulates in the crop. Therefore, there are many research results showing that the productivity and quality of crops are affected by the supplied nitrogen form, and mixing NO_3_^−^ with NH_4_^+^ ions rather than supplying 100% of the nitrogen in the form of NO_3_^−^ is better [4,5,6,7]. As more NO_3_^−^ is supplied, plants are more likely to perish, and their storability and taste decline [8]. In addition, high NO_3_^−^ and NO_2_^−^ contents in eaten vegetables render harm to animals or humans who consume them [9]. In particular, it is bad for children or young animals to eat vegetables with accumulated nitrates. Children are more susceptible to severe NO_3_^−^ poisoning. This is because bacteria that cause nitrate to become toxic in a child’s digestive organs for months after their birth can be parasitic [10]. These bacteria convert NO_3_^−^ into a toxic substance, nitrite ions. Nitrite ions (NO_2_^−^) react with hemoglobin, which serves to transport oxygen in the body, to produce methemoglobin, which does not transport O_2_, as shown via the reaction below [11].
NO_2_^−^ + oxyHb (Fe^2+^) → metHb (Fe^3+^) + NO_3_^−^.

As hemoglobin is substituted with methemoglobin, the amount of oxygen transported in the body decreases [12]. As the amount of oxygen decreases, it gradually becomes more difficult for the child to breathe. This phenomenon is referred to as methemoglobinemia and is generally referred to as the ‘blue baby’ disease. In addition, these substances act as carcinogens through reactions in the body [11].

When plants absorb nitrogen in the form of ammonium, the least energy is used in the process of its conversion to amino acids [13,14]. However, when NH_4_^+^ accumulates in plant cells, the production of free radicals increases, which increases oxidative stress and causes adverse effects [13,15].

Urea has been studied less than previously mentioned nitrate and ammonium ions, but it has been shown that urea, which is also considered an ammonium fertilizer, is properly metabolized in plants [2]. However, some studies have reported that excessive accumulation of urea in plant tissues reduces growth and changes the color of leaves.

When plants absorb nitrogen, the response varies, even in the same species [16]. Different plant species have different expressions of resistant genes to adverse effects of ammonium [17]. Some plants are more sensitive than others. Therefore, different plant species respond differently to ammonium and nitrate in the nutrient solution. Silicon added to the nutrient solution can reduce plant damage caused by the nitrogen source [18]. The action of Si on plants strengthens the cell wall, which improves the leaf structure and the shape [19]. As a result, Si enhances light absorption and increases the photosynthetic efficiency. Silicon increases the antioxidant activity of plants and protects plants from oxidative damage [20,21,22]. As silicon accumulates in plant cell walls, it can protect plant cells from exposure to excess NH_4_^+^ in the medium [16]. Therefore, plant growth may be further promoted by manipulating the ratio of nitrate, ammonium, and urea as the nitrogen source. Excessive tissue nitrate contents violate food safety, while excessive levels of ammonium in tissues are toxic to the plant. To reduce this ammonium phytotoxicity, silicon may be used.

Ssamchoo is one of the leafy vegetables that respond most sensitively to the characteristics of the nutrient solution such as the NO_3_^−^/NH_4_^+^ ratio. Ssamchoo is a hybrid plant developed in Korea by hybridizing Chinese cabbages and regular cabbages [23]. In particular, Ssamchoo has substantial fiber, which is beneficial for digestion, and it contains more vitamin A, iron, ascorbic acid, and calcium than Chinese cabbage, cabbage, and lettuce. It is very important to find a suitable nutrient solution because Ssamchoo is widely used in various dishes in Korea including ssam or wraps and salad dishes. In particular, it was used in this study because it is a leafy vegetable that is very sensitive to growth as affected by the composition of the nutrient solution [23]. Experiments were conducted by adjusting ratios of the nitrogen sources and supplementing silicon to Ssamchoo (*Brassica* lee ssp. *namai* cv. Ssamchoo).

## 2. Results

### 2.1. Selection of the Nutrient Solution

#### 2.1.1. Germination and Early Growth

The MGT was greater with GNU1 than with GNU2 and Peters Professional in Ssamchoo (Table 1).

In an initial growth measurement conducted on sprouts at 7 days after sowing, Peters Professional and GNU2 tended to result in longer shoot lengths and leaf lengths than GNU1 did. However, when observing their appearance, it was confirmed that the leaves turned yellow in Ssamchoo grown in Peters Professional (Figure 1).

#### 2.1.2. Growth and Development

Ssamchoo was harvested 28 days after sowing. When observed from the side, Ssamchoo grown in Peters Professional had low shoot lengths (Figure 2). In Ssamchoo grown in Peters Professional, the leaf part except for the veins turned yellow, and the yellowing degree was more pronounced at the edge of the leaf (Figure 2). When the roots were observed under a microscope, the root color of Ssamchoo grown in Peters Professional was darker than that of other treatments. Ssamchoo grown in Peters Professional showed stopped root hair development, where the root hairs were very short and did not grow sufficiently (Figure 2). Furthermore, Ssamchoo showed significantly longer shoot length, leaf area, and leaf width when grown in GNU1 and GNU2 than in Peters Professional (Figure 3). There were no significant differences in the number of leaves as affected by the treatment in Ssamchoo (Figure 3). On the other hand, when the specific leaf weight was measured, the leaf thickness was greater when grown in Peters Professional (Figure 3). However, the fresh weight was significantly less in Peters Professional than in GNU1 and GNU2 (Figure 3).

The CAT and POD activities in Ssamchoo were significantly higher in GNU1 than in GNU2 and Peters Professional. It appeared that Ssamchoo was healthier when grown in GNU1 than in GNU2 or Peters Professional. However, it was found that the CAT activity was higher in Peters Professional than in GNU2. This seems to be due to the stress caused by adverse effects of ammonium (Figure 4).

When the chlorophyll content was compared, there were no significant differences (Figure 5). There were no significant differences in the vitamin C content of Ssamchoo as affected by the nutrient solution (Figure 5). The NH_4_^+^, NO_2_^−^, and NO_3_^−^ contents in the tissues were analyzed (Figure 6). The results showed that the plants grown in Peters Professional had high tissue NH_4_^+^ content. In Ssamchoo, tissue NO_2_^−^ content was higher in GNU2 than in GNU1 and Peters Professional, and tissue NO_3_^−^ content was higher in GNU1 than in GNU2 and Peters Professional.

### 2.2. Effects of the NO_3_^−^/NH_4_^+^ Ratio in the Nutrient Solution

#### 2.2.1. Germination

As a result of calculating the GE of Ssamchoo, it was found that most of the germination occurred in all treatments 3 days after sowing (Table 2). In terms of GR, it was found that GNU2-4 led to the fastest germination, and the MGT was slightly longer in the GNU1 group than in the GNU2 group.

#### 2.2.2. Growth and Development

Ssamchoo was harvested 28 days after sowing. At harvest, yellow leaves were found in GNU1-1, GNU2-1, GNU1-4, and GNU2-4 (Figure 7).

It was found that the color of the roots was darker in GNU1-4 and GNU2-4, which had the highest NH_4_^+^ content compared to other treatments. The fresh weight was significantly higher in GNU1-2, GNU1-3, GNU2-1, and GNU2-2 than in other treatments. Ssamchoo showed the lowest fresh weight in GNU1-1. The specific leaf weight was the greatest in GNU2-1 (Figure 8). GNU2 resulted in a higher vitamin C tendency than GNU1. The vitamin C content was significantly higher in Ssamchoo in the GNU2-2 treatment, followed by the GNU1-2 treatment (Figure 8).

There were no significant differences between the treatments in the tissue ammonium content (Figure 9). GNU1-1 and GNU2-2 resulted in significantly higher nitrite content. 

### 2.3. Effects of Urea and Silicon

Harvest was performed 28 days after sowing. The appearances were observed after harvest (Figure 10). Treatments GNUb and GNUb’ had smaller leaves than other treatments. Yellowing was observed in some leaves. When observing the roots, it was found that treatments GNUa‘ and GNUd’ had especially high root density (Figure 10). 

Treatments GNUa‘, GNUb’, GNUc’, and GNUd’ led to higher root density and longer root hairs than treatments GNUa, GNUb, GNUc, and GNUd. In other words, silicon treatment acted effectively on the development of roots. It was found that the fresh weight in treatment GNUc’ was significantly higher than that in the other treatments (Figure 11). 

It was very beneficial to increase the fresh weight by properly controlling the ratio of NO_3_^−^, NH_4_^+^, and urea as the source of nitrogen and using them all. Furthermore, it can be seen that the addition of Si had an effect on increasing the fresh weight. There were no significant differences in the specific leaf weight as affected by the treatments (Figure 12). The number of leaves was significantly higher in treatment GNUc than in the other treatments. The leaf length and leaf width were significantly higher in treatments GNUc and GNUc’ than in the other treatments (Figure 12).

This shows that the nitrogen source greatly affected the leaf development. In addition, treatment GNUc’ showed significantly higher leaf length and leaf width than treatment GNUc, indicating that Si also had a positive effect on leaf development in this treatment. However, Si did not affect the leaf development in treatments GNUa and GNUb. In treatments GNUc, and GNUd, in which urea was added to the nitrogen source, the Si supply affected the size of the leaves. The chlorophyll content was significantly higher in treatments GNUa and GNUc’ than in other treatments (Figure 13). Root activities were higher in treatments GNUa‘, GNUb’, GNUc’, and GNUd’ than in treatments GNUa, GNUb, GNUc, and GNUd (Figure 13). This shows that the Si supply had a significant effect on the root development, as well as the root activities.

## 3. Discussion

The nutrient solution is one of the factors that has the greatest influence on plant growth and development [24]. Therefore, the development of a nutrient solution for home hydroponic systems is of utmost importance. In particular, since Ssamchoo reacted sensitively to the nitrogen source of the nutrient solution, the aim was to find a nutrient solution suitable for the growth and development of Ssamchoo.

First, three nutritional solutions, GNU1, GNU2, and Peters Professional, were supplied to Ssamchoo. In particular, among the nutritional solutions, in the Ssamchoo supplied with Peters Professional, the leaves were yellow, or the roots were dark, and the fresh weight was significantly lower. This result is reflective of the typical adverse effects of ammonium [11]. Adverse effects of ammonium in plants typically result in reduced growth and yellowing of leaves [15]. In particular, it greatly affects the development of roots. Adverse effects of ammonium lead to little development of the root hairs, darkening of color, and serious morphological and physiological changes [25]. High NH_4_^+^ produces stunted roots, while high NO_3_^−^ increases root growth [6,12]. Nitrate is an essential ingredient in crops. In general, what is absorbed from the root is almost entirely consumed in the leaves, leaving little to be accumulated. Through Experiment 1, it was confirmed that the commercial nutrient solution, which was considered suitable for leafy vegetables, was not suitable for Ssamchoo. Thus, in the second experiment, Peters Professional, which caused adverse effects of ammonium in Ssamchoo, was excluded. Instead, the treatment solutions were subdivided according to the ratio of nitrogen source (NO_3_^−^/NH_4_^+^) of GNU1 and GNU2. When the MGTs were compared, the GNU2 group showed a faster tendency than the GNU1 group, but the ratio of N had no effect on the germination. The Ssamchoo nitrate content was significantly higher than in other treatments. Nitrate ions supplied in the nutrient solution affect the concentration of nitrate ions in the leaf tissue. In Ssamchoo, the concentration of nitrate ions was the highest in GNU1-1, a treatment with a high nitrate ion concentration in the GNU1 group, and, as the concentration of nitrate ions in the nutrient solution decreased, the nitrate ion content in the tissues tended to decrease. However, there was no such trend in the GNU2 group. The reason why the tendency did not appear seems to be that the inorganic elements necessary for the growth of Ssamchoo in the GNU2 group were insufficient to accumulate inside the tissues. In addition, the fact that GNU2-3 and GNU2-4 showed higher nitrate contents than GNU2-1 and GNU2-2 seems to be a ‘concentrated effect’ because the fresh weight was relatively small [26]. Moreover, the type of nitrogen source can affect the synthesis of hormones in the roots by inhibiting or promoting cell elongation [27]. A high proportion of NO_3_^−^ promotes the synthesis of hormones and transport of auxins, while a high proportion of NH_4_^+^ can inhibit the activity of these hormones.

The toxic mechanism of ammonium ions in plants is complex and only partially known. Several studies have shown that the content of ammonium ions is correlated with the content of putrescine in plants [28]. Several studies have shown that the accumulation of putrescine negatively affects plant growth and development, causing events such as potassium leakage, protein loss, membrane depolarization, and tissue necrosis. These results showed that the combination of nitrate, ammonium, and urea in the nutrient solution is a way to reduce the accumulation of putrescine. When three nitrogen sources were supplied at the same time from the nutrient solution to wheat, the absorption rate of nitrogen increased, and growth was improved [29].

The nutrient solution with an 83.3:16.7 NO_3_^−^/NH_4_^+^ ratio, which gave the greatest fresh weight in the second experiment, was chosen as the control, while the solution with a 50:50 NO_3_^−^/NH_4_^+^ ratio with a lower nitrate content among the two unfavorable treatments was selected as a treatment group in the third experiment. A part of NH_4_^+^ was replaced with urea to make four different NO_3_^−^/NH_4_^+^/urea ratios (83:17:0, 50:50:0, 50:25:25, and 50:0:50) in combination with two levels of Si (0 and 10.7 mmol·L^−1^ Si) supplemented to evaluate its alleviating effect of ammonium’s adverse effects. If there are more nitrate ions in the nutrient solution than necessary, they are absorbed excessively, and the remaining amount accumulates in the crop [28]. Then, the crop is prone to decay, which makes it less storable, tasteless, and bad for human and animal health [9,12,30]. Thus, the form of nitrogen contained in the nutrient solution is directly linked to various results, such as food safety, plant growth and development, and phytotoxicity in the plant. Therefore, the nitrogen source must be properly mixed or balanced [2,28]. In particular, it is bad for children to eat vegetables with accumulated nitrates, since they are susceptible to severe NO_3_^−^ poisoning [30].

Therefore, in Experiment 3, the solution with a NO_3_^−^/NH_4_^+^ ratio of 50:50 was selected as the experimental group, aimed at improving the problem of adverse effects of ammonium by replacing a part of ammonium with urea and supplementing Si. When the third experiment was conducted, adverse effects of ammonium was more pronounced, especially in the 50:50:0 NO_3_^−^/NH_4_^+^/urea treatment. In the 50:25:25 NO_3_^−^/NH_4_^+^/urea treatment, the root was not overcrowded, and the shoot fresh weight increased 2.28-fold as compared to the 50:50:0 NO_3_^−^/NH_4_^+^/urea treatment. When a portion of the ammonium supply was replaced with urea, the adverse effects of ammonium were significantly reduced. Studies on the mitigation of ammonium toxicity with the supply of silicon have been reported for broccoli [31], yellow passionfruit seedling [32], sugar beet [33], and radish seedlings [27]. Silicon is known to participate in defense reactions, especially in stressful situations. It increases the activity of antioxidant enzymes in plant cells and reduces ROS in tissues. Further studies are needed on the detailed mechanism [22], as there are only few studies on whether the adverse effects of ammonium of Ssamchoo are improved with silicon supplementation. In Experiment 3, the effect of silicon was variable depending on the ratio of ammonium as the source of the nitrogen in the nutrient solution. In the case of the greatest (50:50:0 which had severe adverse effects of ammonium) and low (83.3:16.7:0.0 and 50:0:50, which had no adverse effects of ammonium) NH_4_^+^ ratio treatments, the adverse effects of ammonium were not alleviated by supplemented silicon. However, in the 50:25:25 NO_3_^−^/NH_4_^+^/urea treatment in which a part of ammonium was replaced with urea, the effect of silicon was obvious and significant. In the 50:25:25 NO_3_^−^/NH_4_^+^/urea treatment, it was confirmed that the shoot fresh weight increased 1.5-fold along with increased root length due to the supplementation of silicon. In a previous study that investigated the effects of silicon on ammonium toxicity in cauliflower (*Brassica oleracea* var. *botrytis*) and broccoli (*Brassica oleracea* var. *italica*), it was reduced with silicon supplementation [31]. It was also observed that the leaf area of cauliflower and broccoli increased, electrolyte leakage in cells decreased, and water use efficiency increased with supplementation of silicon. The effects of silicon on adverse effects of ammonium have been revealed in several other plants, such as yellow passionfruit seedlings (*Passiflora edulis* Sims f. *flavicarpa* O. Deg.), sugar beet (*Beta vulgaris* L.), and radish seedlings (*Raphanus sativus* L.) [19,27,32]. In particular, the number of leaves did not decrease, and the leaf area increased due to the addition of silicon. This is very important and meaningful in terms of productivity and usability because Ssamchoo is mainly used for ssam dishes in Korea [34].

## 4. Materials and Methods

### 4.1. Home Hydroponic System

The hydroponic system used in this study was a home hydroponic system in the process of development for commercialization (Tiiun L061G1, LG Electronics Inc., Seoul, Korea). The external dimensions (W × H × D) were 595 mm × 815 mm × 590 mm, and each weighed 68 kg. The interior consisted of two tiers, and each tier was again divided into three compartments. One unit had a total of six compartments. Each compartment could be combined with a pod developed to specifically fit in the system. The nutrient solution was supplied by an ebb-and-flow system, and there was a 3 L water tank at the bottom of the unit. The supply of the nutrient solution intersected the two tiers. The temperature and photoperiod were set at 25/18 °C and 14/10 h day/night, respectively. Light was provided by white LEDs (mainly distributed in red (32.1%), green (43.7%), and blue (19.8%) lights), with a total of 300 µmol·m^−^^2^·s^−^^1^ photosynthetic photon flux density (PPFD), to both the upper and lower tiers. A fan was located in the back at the upper right corner of the unit. The fan airspeed was set at 1.0 m·s^−^^1^ using a ventilation fan with a radius of 38 mm. The cross-sectional area of the tube through which air flows was 0.00453 m^2^. The total volumetric airflow rate was 16.31 m^3^·h^−^^1^.

### 4.2. Pod (Seed Kit)

The dimensions (W × H × D) of the pod (LPH02, LG Electronics Inc., Seoul, Korea) were 119 mm × 35 mm × 339 mm, and the inside was empty. It was a combination of a plastic body and a cover. The interior was divided into a total of 10 cells and was designed to hold a growing medium of 22 mm × 38 mm × 22 mm dimension (W × H × D) in each cell. The bottom of the pod had six elongated holes, and water could enter and exit through these holes at the bottom surface. The sowing was performed first in the growing medium fit into the pod. The sown pods were placed in slots in the hydroponic system. Since the pod has a structure which makes it difficult to separate or replace the growing medium after assembly, three seeds per hole were sown to secure at least one germinated seed per cell, and any extra seedlings were thinned afterward. For seed germination and seedling emergence, holes were checked daily. The germination was determined based on the visual observation of germinated seeds in the inside of the planting holes of the growing medium, and the emergence was counted on the basis of the visual observation of the surface of the growing medium.

### 4.3. Plant Materials

In this experiment, Ssamchoo (*Brassica* lee ssp. *namai* cv. Ssamchoo ‘Chunssamhwang 51′, Asia Seed Co., Ltd., Seoul, Korea) was selected as the test plant. The number of plants used was 30 per treatment, and 10 plants per pod were grouped as a replicate. Germination was observed from the day after sowing. Seeds were counted as germinated when the root radicle broke through the seed coat. Seven days after sowing, extra seedlings were thinned out, leaving only one seedling for cultivation until harvest 28 days after sowing.

### 4.4. Nutrient Solution

In the first experiment, a commercial nutrient solution used was prepared using Peters Professional (20-20-20, The Scotts Co., Marysville, OH, USA), and a multipurpose nutrient solution formulated in Gyeongsang National University was prepared for comparison [35]. The macro-elements and their concentrations for the multipurpose nutrient solution were (in mmol·L^−^^1^) 13.0 N, 12.0 Ca^2+^, 4.0 Mg^2+^, 5.0 K^+^, 4.0 SO_4_^2−^, and 2.0 H_2_PO_4_^−^. The microelements and their concentrations in the multipurpose nutrient solution were (in μmol·L^−^^1^) 20 B, 0.5 Cu, 10 Fe, 10 Mn, 0.5 Mo, and 4 Zn (Table 3). The nutrient solution with the above composition was named GNU1. GNU2 was a nutrient solution with 77% concentration levels of all elements in GNU1.

In the second experiment, eight treatment solutions used, constituting GNU1 and GNU2 in combination with four different of NO_3_**^−^**/NH_4_^+^ ratios (100:0, 83.3:16.7, 66.7:33.3 and 50:50), and they were classified into subgroups as 1, 2, 3, and 4, respectively (Table 4).

In the third experiment, NH_4_^+^ was partially replaced with urea to make four levels of NO_3_^−^/NH_4_^+^/urea ratio (GNUa, 83:17:0; GNUb, 50:50:0; GNUc, 50:25:25; GNUd, 50:0:50), in combination with either 0 or 10.7 mmol·L^−^^1^ Si supplied by K_2_SiO_3_ (Table 5). The silicon supply concentration was determined by referring to previous experiments in our laboratory [35,36]. In this case, the prime symbol (**′**) was added to each code in the treatment to which silicon was added.

Initially, 3 L of each nutrient solution was supplied to each cultivation system. An additional 2 L of each nutrient solution was supplied when the solution level in the tank dropped to below 1 L. The remaining solution in the nutrient tank was taken once a day for the pH and EC measurements. When the solution pH exceeded the set valid range (5.6–6.4), it was adjusted to 6.0.

### 4.5. Germination

After sowing, the insides of sowing holes made in the growing medium were individually observed. The seed was considered germinated when the root radicle broke through the seed coat. Three seeds were sown in each hole, but the germination data were collected for holes with at least one germinated seed. For each treatment, 30 holes were observed for the Ssamchoo. Germination rate (GR), mean germination time (MGT), and germination energy (GE) were calculated from the recorded germination data [33,37,38]. The GR, MGT, and GE were calculated as follows:$$\mathrm{GR}\;\left(\mathrm{seed}\;\mathrm{per}\;\mathrm{day}\right)==\frac{X_{1}}{Y_{1}}+\frac{X_{2}}{Y_{2}}+\ldots +\frac{X_{n}}{Y_{n}}$$
$$\mathrm{MGT}\;\left(\mathrm{day}\right)=\frac{\Sigma \left(X_{n}\times {\;Y}_{n}\right)}{T_{\mathrm{seed}}}$$
$${\mathrm{GE}}_{n}\left(\%\right)=\frac{X_{n}}{T_{\mathrm{seed}}}\times 100$$
where X_1_, X_2_, and X_n_ are the number of seeds (holes) germinated on the first, second, and n-th days, respectively, and Y_1_, Y_2_, and Y_n_ are the number of days from sowing to first, second, and n-th counts, respectively. T_seed_ is the total number of seeds (holes) sown, and GE_n_ is germination energy on day ‘n’ after sowing.

### 4.6. Measurement of Growth Parameters

The shoot length was measured from the crown to the highest point of the shoot. As for the leaf number, all the leaves generated per plant were counted, but cotyledons were excluded. The leaf length and leaf width of the largest leaf on each individual plant were measured. The fresh weight was measured for each plant, but only the weight of the shoot was measured. To measure the specific leaf weight, 2 cm^2^ leaf discs were taken from the youngest mature leaf using a cork borer, and the specific leaf weight was calculated as follows:Specific leaf weight (g·cm^−2^) = leaf weight (g)/ leaf area (cm^2^).

The moisture content of the leaves was calculated after measuring the fresh weight and dry weight of the leaves. At 7 days after sowing, all leaves were collected from three individual plants per each treatment. At 28 days after sowing, three fully grown leaves were collected from three individual plants per treatment, and the fresh weight was measured immediately after the harvest. The samples were then dried in a dry oven (Model FO-450M, Jeio Technology Co., Ltd., Seoul, Korea) at 60 °C for 48 h to measure the dry weight and moisture content of the leaves.

### 4.7. Chlorophyll Content and Root Activity

Samples for the chlorophyll content measurements were taken in the same way as for the specific leaf weight. Ethanol, acetone, and distilled water were mixed to make an extraction solution [39]. The collected sample was immersed in this solution and the chlorophyll was extracted at 4 °C for 24 h. The absorbance of the extracted solution was measured at 645 and 663 nm using a spectrometer (Libra S22, Biochrom Co., Ltd., Cambridge, UK), and the chlorophyll content was subsequently calculated according to the method described by Noh and Jeong (2021) [40]. Root samples (0.1 g) were collected to measure the root activities. A 1:1 mixed solution of 0.4% tetrazolium chloride and 0.2 M Tris-HCl was prepared. Root samples were placed in this solution and incubated for 16 h, while light was blocked. The root samples that changed color were then taken out, and the solution on the surface was blot-dried. After 24 h of immersion in a 99% methanol solution, the absorbance was measured at 485 nm using a spectrometer (Libra S22, Biochrom Co., Ltd., Cambridge, UK) [41].

### 4.8. Contents of Soluble Sugar and Starch

The contents of starch and soluble sugars were determined using the Anthrone colorimetric method according to Vasseur et al. (2011) [42] and Ren et al. (2018) [43]. For the extract, 0.3 g of the frozen sample was pulverized, to which 10 mL of distilled water was added and incubated in water at 100 °C for 30 min; then, 15 mL of distilled water was subsequently added to make the total solution volume 25 mL. This was then incubated in 100 °C water for 30 min. When the temperature dropped to room temperature, the extract was centrifuged (5430R, Eppendorf, Hamburg, Germany) at 6500 rpm for 10 min. Following this, 2.5 mL of the supernatant was separated, and distilled water was added to make the total solution volume 10 mL. A 0.2 mL sample was taken from this solution, to which 1.8 mL of distilled water was added. Next, a mixture of 0.5 mL of 98% anthrone (C_14_H_10_O) and ethyl acetate (CH_3_COOC_2_H_5_), along with 5 mL of 98% sulfuric acid, was added to the total solution, shaken well, and again incubated at 100 °C for 10 min. After the reaction finished and the temperature dropped to room temperature, the absorbance was measured at 630 nm with a spectrometer (Libra S22, Biochrom Co., Ltd., Cambridge, UK). The starch content was determined using the solids remaining after preparing the extract for measuring the soluble sugar contents. The solids were collected, 20 mL of distilled water was added, and 2 mL of HClO_4_ was added to make a total solution of 22 mL. The total solution was incubated for 30 min in 100 °C water and filtered through a filter. A 0.5 mL sample was obtained from the extract and 1.5 mL of distilled water was added. After adding 1 mL of 98% anthrone and ethyl acetate mixed solution, 5 mL of 98% sulfuric acid was added. After the reaction sufficiently occurred, the mixture was incubated for 30 min until the temperature dropped to room temperature. The absorbance of the extracted solution was measured at 485 nm using a spectrometer (Libra S22, Biochrom Co., Ltd., Cambridge, UK).

### 4.9. Soluble Proteins and Activities of Antioxidant Enzymes

The total protein estimations were conducted using Bradford’s reagent [14,44]. The superoxide dismutase (SOD) activity was estimated by following the nitro blue tetrazolium (NBT) inhibition methods according to the protocols of Giannopolitis and Ries (1977) [45]. The activity of the catalase (CAT) enzyme was measured according to the method of Cakmak and Marschner (1992) [46]. The peroxidase (POD) activity was determined on the basis of the amount of enzyme required for the formation of tetra guaiacol per minute, following the methods of Shah et al. (2001) [47]. The ascorbate peroxidase (APX) activity was assayed following the methods of Nakano and Asada (1981) [48].

### 4.10. Statistical Analysis

The SAS statistical software (Version 9.2, SAS Inst., Cary, NC, USA) was used for the statistical analysis of variance (ANOVA) and Duncan’s multiple range test at a significance level of *p* = 0.05.

## 5. Conclusions

In conclusion, the ratio of different nitrogen sources is an important nutrient solution factor in home hydroponic systems. Adding silicon to a nutrient solution containing NO_3_^−^, NH_4_^+^, and urea was the most effective approach to increase the fresh weight. The number of leaves and leaf area are important indicators of high productivity since the Ssamchoo is used in ssam dishes. It can be concluded that a solution with a NO_3_^−^/NH_4_^+^/urea ratio of 50:25:25, along with the supplementation of 10.7 mmol·L^−1^ Si, is the most suitable nutrient solution for growing Ssamchoo in the newly developed home hydroponic system using light-emitting diodes (LEDs) as the sole source of light.

## Figures and Tables

**Figure 1 plants-11-02882-f001:**
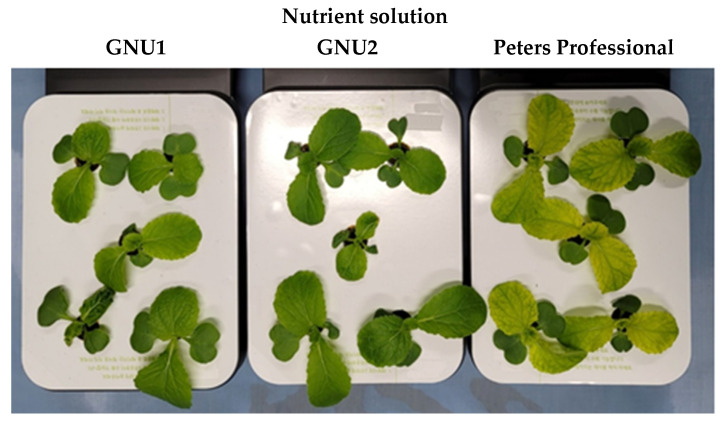
Growth of Ssamchoo as affected by the nutrient solution 7 days after sowing. Peters Professional, GNU1, and GNU2 were the nutrient solutions used in the experiment. Peters Professional is a commercial nutrient solution.

**Figure 2 plants-11-02882-f002:**
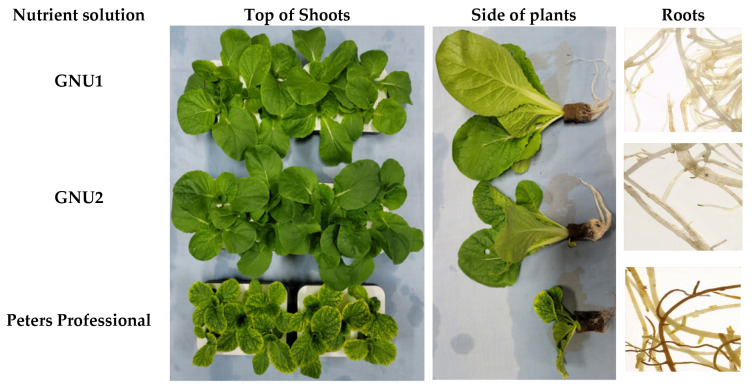
Growth of Ssamchoo as affected by nutrients 28 days after sowing in Experiment 1. Peters Professional, GNU1, and GNU2 were the nutrient solutions used in the experiment. Peters Professional is a commercial nutrient solution.

**Figure 3 plants-11-02882-f003:**
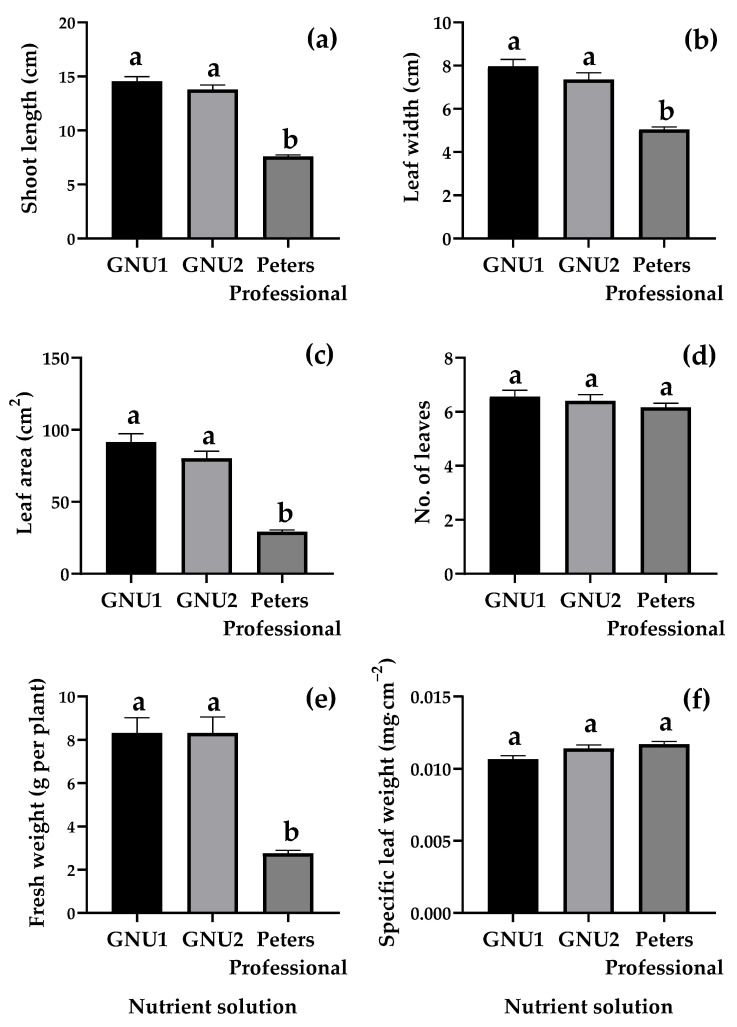
Shoot length (**a**), leaf width (**b**), leaf area (**c**), number of leaves (**d**), fresh weight (**e**), and specific leaf weight (**f**) of Ssamchoo as affected by the nutrient solution 28 days after sowing in Experiment 1. The vertical bars represent the SEs of 30 biological replicates (*n* = 30). Significant differences among treatments are indicated by lowercase letters at *p* ≤ 0.05 according to Duncan’s multiple range test.

**Figure 4 plants-11-02882-f004:**
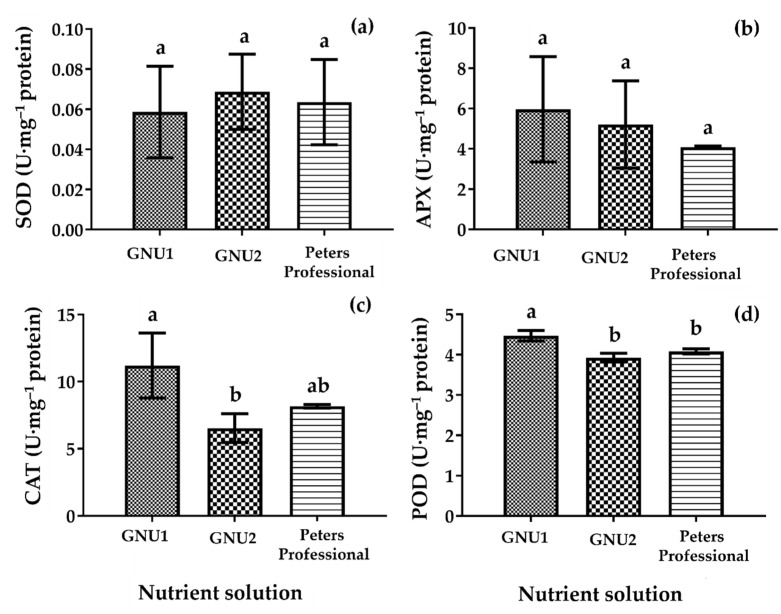
Activities of antioxidant enzymes SOD (**a**), APX (**b**), CAT (**c**), and POD (**d**) in Ssamchoo as affected by the nutrient solution at 28 days after sowing in Experiment 1. Peters Professional, GNU1, and GNU2 were the nutrient solutions used in the experiment. Peters Professional is a commercial nutrient solution. Vertical bars represent the SEs of three biological replicates (*n* = 3). Significant differences among treatments are indicated by lowercase letters at *p* ≤ 0.05 according to the Duncan’s multiple range test, and different lowercase letters indicate significant differences.

**Figure 5 plants-11-02882-f005:**
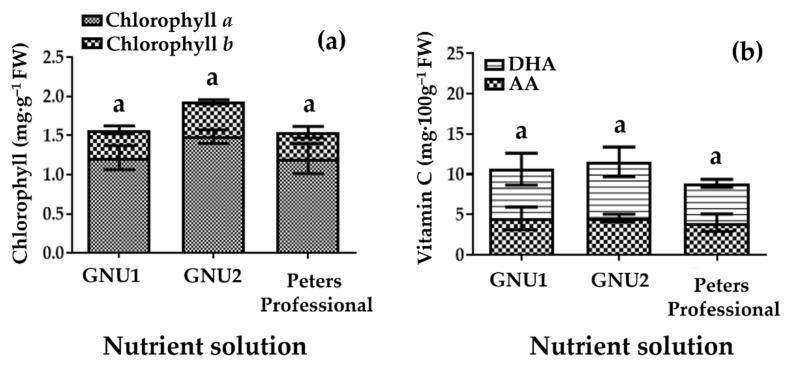
Contents of chlorophyll a and b (**a**) and contents of ascorbic acid (AA) and L-dehydroascorbic acid (DHA) (**b**) in Ssamchoo as affected by the nutrient solution 28 days after sowing in Experiment 1. Since ascorbic acid is easily oxidized into L-dehydroascorbic acid, vitamin C was calculated as the total ascorbic acid (TAA), a sum of AA and DHA. Peters Professional, GNU1, and GNU2 were the nutrient solutions used in the experiment. Peters Professional is a commercial nutrient solution. Vertical bars represent the SEs of three biological replicates (*n* = 3). Significant differences among treatments are indicated by lowercase letters at *p* ≤ 0.05 according to the Duncan’s multiple range test.

**Figure 6 plants-11-02882-f006:**
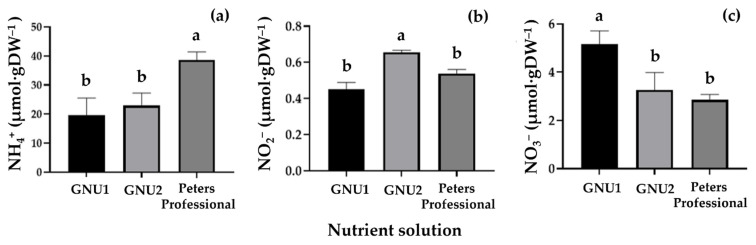
Tissue contents of NH_4_^+^ (**a**), NO_2_^−^ (**b**), and NO_3_^−^ (**c**) in the leaves of Ssamchoo as affected by the nutrient solution 28 days after sowing in the experiment 1. Vertical bars represent the SEs of three biological replicates (*n* = 3). Significant differences among treatments are indicated by lowercase letters at *p* ≤ 0.05 according to the Duncan’s multiple range test, and different lowercase letters indicate significant differences.

**Figure 7 plants-11-02882-f007:**
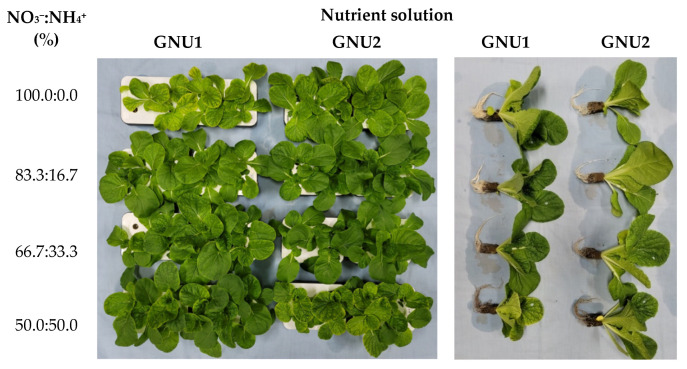
Growth of Ssamchoo as affected by the nutrient solution 28 days after sowing in Experiment 2. Eight treatment solutions were GNU1 and GNU2 in combination with four levels of NO_3_^−^/NH_4_^+^ ratio (100:0, 83.3:16.7, 66.7:33.3, and 50:50), and they were classified into subgroups as 1, 2, 3, and 4, respectively.

**Figure 8 plants-11-02882-f008:**
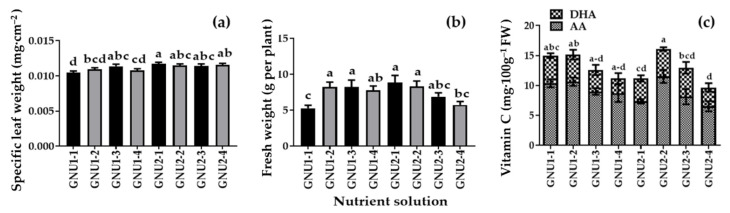
Specific leaf weight (**a**), fresh weight (**b**), and contents of ascorbic acid (AA) and L-dehydroascorbic acid (DHA) (**c**) of Ssamchoo as affected by the nutrient solutions 28 days after sowing in the experiment 2. The vertical bars represent the SEs of 30 biological replicates (*n* = 30). Significant differences among treatments are indicated by lowercase letters at *p* ≤ 0.05 according to Duncan’s multiple range test.

**Figure 9 plants-11-02882-f009:**
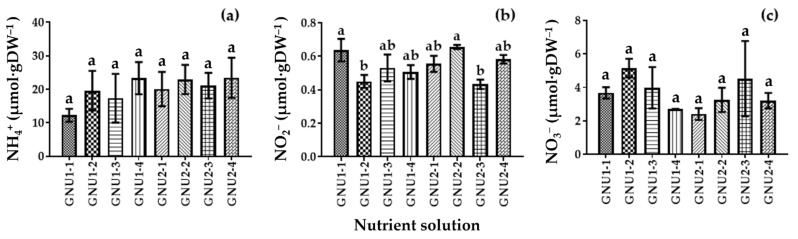
Tissue contents of NH_4_^+^ (**a**), NO_2_^−^ (**b**), and NO_3_^−^ (**c**) as affected by the nutrient solution in Experiment 2. Vertical bars represent the SEs of three biological replicates (*n* = 3). Significant differences among treatments are indicated by lowercase letters at *p* ≤ 0.05 according to Duncan’s multiple range test.

**Figure 10 plants-11-02882-f010:**
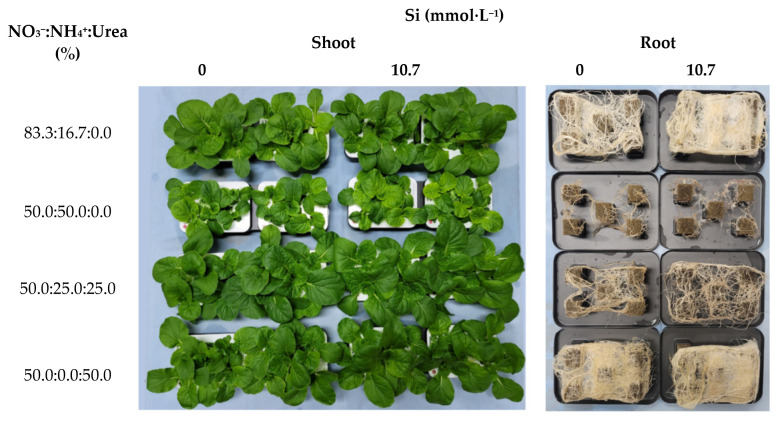
Shoot and root growth of Ssamchoo as affected by the nutrient solution 28 days after sowing in Experiment 3. The eight treatment solutions were four levels of NO_3_^−^/NH_4_^+^/urea ratio (83:17:0, 50:50:0, 50:25:25, and 50:0:50) in combination with two levels of Si (0 and 10.7 mmol·L^−1^ Si).

**Figure 11 plants-11-02882-f011:**
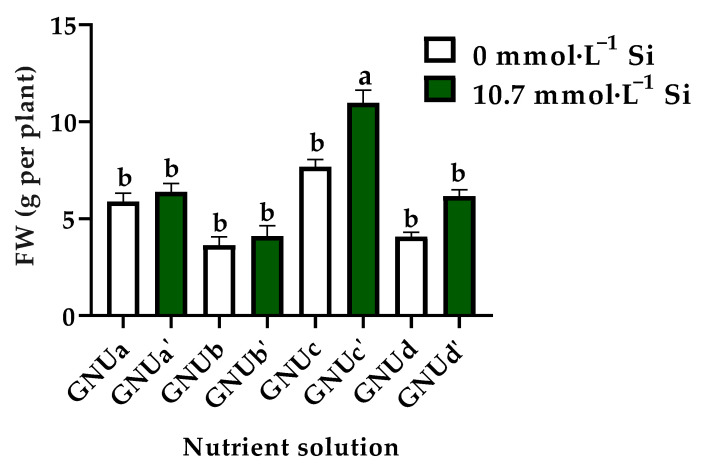
Fresh weight of Ssamchoo as affected by the nutrient solution 28 days after sowing in Experiment 3. Nutrient solutions had one of four levels of NO_3_^−^/NH_4_^+^/urea ratio (GNUa and GNUa‘, 83:17:0; GNUb and GNUb’, 50:50:0; GNUc and GNUc’, 50:25:25; GNUd and GNUd’, 50:0:50) in combination with either 0 (GNUa, GNUb, GNUc, and GNUd) or 10.7 mmol·L^−1^ Si (GNUa‘, GNUb’, GNUc’, and GNUd’). Vertical bars represent the SEs of three biological replicates (*n* = 3). Significant differences among treatments are indicated by lowercase letters at *p* ≤ 0.05 according to Duncan’s multiple range test.

**Figure 12 plants-11-02882-f012:**
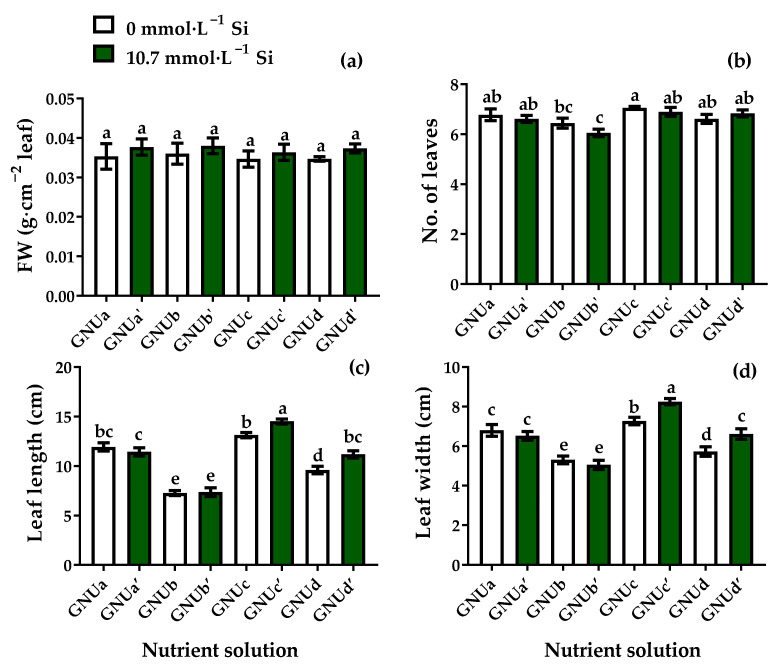
Specific leaf weight (**a**), number of leaves (**b**), leaf length (**c**), and leaf width (**d**) of Ssamchoo as affected by the nutrient solution in Experiment 3. Vertical bars represent the SEs of three biological replicates (*n* = 3). Significant differences among treatments are indicated by lowercase letters at *p* ≤ 0.05 according to Duncan’s multiple range test.

**Figure 13 plants-11-02882-f013:**
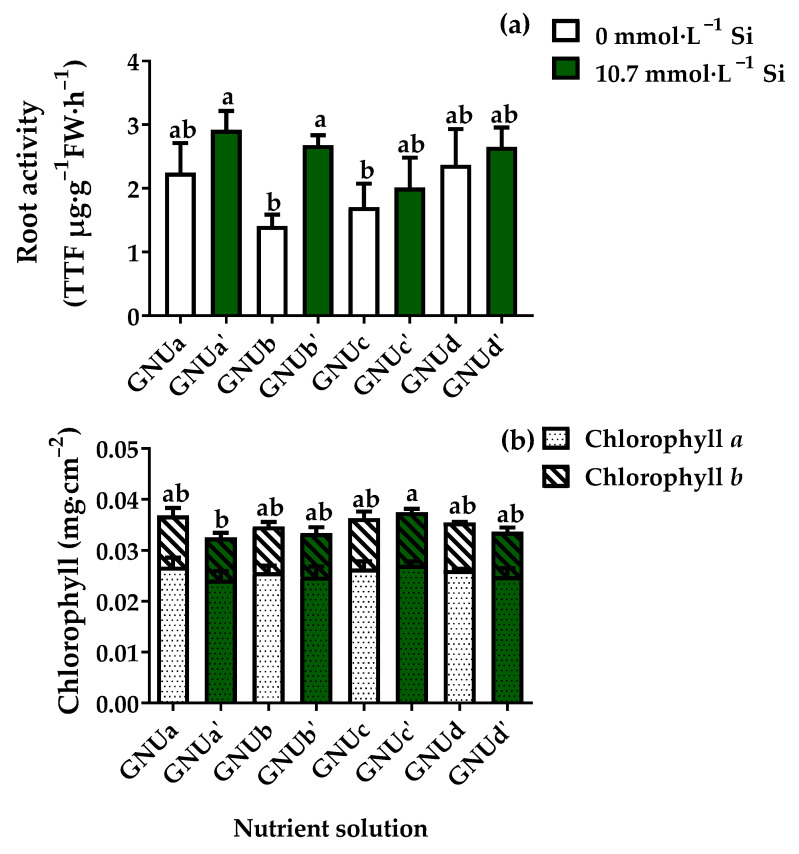
Root activity (**a**) and contents of chlorophylls (**b**) in Ssamchoo as affected by the nutrient solution in Experiment 3. Nutrient solutions had one of four levels of NO_3_^−^/NH_4_^+^/urea ratio (GNUa and GNUa‘, 83:17:0; GNUb and GNUb’, 50:50:0; GNUc and GNUc’, 50:25:25; GNUd and GNUd’, 50:0:50) in combination with either 0 (GNUa, GNUb, GNUc, and GNUd) or 10.7 mmol·L^−1^ Si (GNUa‘, GNUb’, GNUc’, and GNUd’). Vertical bars in root activity represent the SEs of the three biological replicates (*n* = 3). Significant differences in root activity among treatments are indicated by lowercase letters at *p* ≤ 0.05 according to Duncan’s multiple range test. Vertical bars in chlorophyll contents represent the SEs of the three biological replicates (*n* = 3) of chlorophyll a and b. Significant differences in total chlorophyll content among treatments are indicated by lowercase letters at *p* ≤ 0.05 according to Duncan’s multiple range test.

**Table 1 plants-11-02882-t001:** Germination rate (GR), mean germination time (MGT), and germination energy (GE) of Ssamchoo as affected by nutrient solutions in the experiment 1.

Nutrient Solution	GR ^z^ (Seeds per Day)	MGT (Day)	GE_n_ (%)	
			GE_2_	GE_3_
GNU1	7.8	2.04 a ^y^	6.7	93.3 b
GNU2	8.0	1.97 ab	6.7	100.0 a
Peters Professional	7.8	1.97 ab	10.0	100.0 a

^z^ GR, germination rate; MGT, mean germination time; GEn, germination energy on day ‘n’ after sowing. ^y^ Significant differences calculated using the Duncan’s multiple range test at *p* ≤ 0.05, and different lowercase letters indicate significant differences.

**Table 2 plants-11-02882-t002:** Germination rate (GR), mean germination time (MGT), and germination energy (GE) of Ssamchoo as affected by the nutrient solution in the experiment 2.

Nutrient Solution (A)	NO_3_^−^/NH_4_^+^ (%) (B)	Code	GR ^y^ (Seeds per Day)	MGT (Day)	GE (%)	
					GE_2_	GE_3_
GNU1	100.0:0.0 ^z^	GNU1-1	8.1 bc ^w^	1.97 ab	10.0 b	93.3
	83.3:16.7	GNU1-2	7.8 c	2.00 a	6.7 b	93.3
	66.7:33.3	GNU1-3	8.8 b	1.84 c	16.7 b	100.0
	50.0:50.0	GNU1-4	8.5 bc	1.87 bc	13.3 b	100.0
GNU2	100.0:0.0	GNU2-1	8.3 bc	1.90 abc	10.0 b	100.0
	83.3:16.7	GNU2-2	8.0 bc	1.93 abc	6.7 b	100.0
	66.7:33.3	GNU2-3	8.8 b	1.84 c	16.7 b	100.0
	50.0:50.0	GNU2-4	10.0 a	1.67 d	33.3 a	100.0
F-test ^x^	A		*	*	NS	NS
	B		*	*	*	NS
	A × B		*	NS	*	NS

^z^ Eight treatment solutions were GNU1 and GNU2 in combination with four levels of NO_3_^−^/NH_4_^+^ ratio (100:0, 83.3:16.7, 66.7:33.3, and 50:50), and they classified into subgroups as 1, 2, 3, and 4, respectively. ^y^ GR, germination rate; MGT, mean germination time; GE_n_, germination energy on day ‘n’ after sowing. ^x^ NS and * represent not significant or significant differences, respectively, at *p* = 0.05, 0.01, or 0.001. ^w^ Significant differences calculated by Duncan’s multiple range test at *p* ≤ 0.05.

**Table 3 plants-11-02882-t003:** Concentrations of macro-element cations and anions used in nutrient solutions in Experiment 1.

Nutrient Solution	Ion Concentration (me·L^−1^)								
	Cation					Anion			Total
	H^+^	Ca^2+^	Mg^2+^	K^+^	NH_4_^+^	NO_3_^−^	SO_4_^2−^	H_2_PO_4_^−^	
GNU1	0.0	6.0	2.0	5.0	2.2	10.8	2.4	2.0	30.4
GNU2	0.0	5.0	1.0	4.0	1.7	8.3	2.4	1.0	23.4

**Table 4 plants-11-02882-t004:** Concentrations of macro-element cations and anions used in nutrient solutions in experiment 2.

Nutrient Solution	NO_3_^−^/NH_4_^+^ (%)	Ion Concentration (me·L^−1^)								
		Cation					Anion			Total
		H^+^	Ca^2+^	Mg^2+^	K^+^	NH_4_^+^	NO_3_^−^	SO_4_^2−^	H_2_PO_4_^−^	
GNU1-1	100.0:0.0	0.0	7.9	2.6	6.5	0.0	13.0	2.0	2.0	34.0
GNU1-2	83.3:16.7	0.0	6.0	2.0	5.0	2.2	10.8	2.4	2.0	30.4
GNU1-3	66.7:33.3	0.0	6.0	2.0	5.0	4.3	8.7	6.6	2.0	34.6
GNU1-4	50.0:50.0	0.0	6.0	2.0	5.0	6.5	6.5	11.0	2.0	39.0
GNU2-1	100.0:0.0	0.0	6.0	1.2	4.8	0.0	10.0	1.0	1.0	24.0
GNU2-2	83.3:16.7	0.0	5.0	1.0	4.0	1.7	8.3	2.4	1.0	23.4
GNU2-3	66.7:33.3	0.0	5.0	1.0	4.0	3.3	6.7	5.6	1.0	26.6
GNU2-4	50.0:50.0	0.0	5.0	1.0	4.0	5.0	5.0	9.0	1.0	30.0

**Table 5 plants-11-02882-t005:** Concentrations of macro-element cations and anions used in nutrient solutions in experiment 3.

Nutrient Solution	NO_3_^−^/NH_4_^+^/Urea (%)	Si (mmol·L^−1^)	Ion Concentration (me·L^−1^)								
			Cation					Anion			Total
			H^+^	Ca^2+^	Mg^2+^	K^+^	NH_4_^+^	NO_3_^−^	SO_4_^2−^	H_2_PO_4_^−^	
GNUa	83:17:0	0.0	0.0	6.0	2.0	5.0	2.2	10.8	2.4	2.0	30.4
GNUa’	83:17:0	10.7	1.4	6.0	2.0	5.0	2.0	11.0	2.0	2.0	30.0
GNUb	50:50:0	0.0	0.0	6.0	2.0	5.0	6.5	6.5	11.0	2.0	39.0
GNUb’	50:50:0	10.7	0.0	6.0	2.0	5.0	6.5	6.5	9.6	2.0	37.6
GNUc	50:25:25	0.0	0.0	6.0	2.0	5.0	6.5	6.5	7.8	2.0	35.8
GNUc’	50:25:25	10.7	0.0	6.0	2.0	6.4	6.5	6.5	7.8	2.0	37.2
GNUd	50:0:50	0.0	0.0	6.0	2.0	5.0	6.5	6.5	4.5	2.0	32.5
GNUd’	50:0:50	10.7	0.0	6.0	2.0	5.0	6.5	6.5	3.1	2.0	31.1

## Data Availability

Not applicable.

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
