# Peer review of "Silicon Supplementation Alleviates Adverse Effects of Ammonium on Ssamchoo Grown in Home Cultivation System"

_plants, 2022, doi:10.3390/plants11212882_

Round 1
Reviewer 1 Report
The manuscript “Alleviation of ammonium toxicity and suppressed growth in Ssamchoo grown in home cultivation system by silicon supplementation to nutrient solution” highlights the beneficial impact of ammonium and nitrate nutrition in the proper ratio for plant health and the role of Si in mitigating ammonium toxicity. Overall, the manuscript is well written but I have a few questions before it can proceed further.
1- The silicon's role is unclear; it reduces ammonium toxicity, but the exact mechanism (any signaling pathway) is missing throughout.
2- Abstract needs to be checked again. Line 30: Through this experiment, GNU1-2 was the most advantageous for the productivity of Ssamchoo. It is not clear here whether GNU1 or GNU 2 is a better cultivar.
3- Authors have used Si to counter ammonia toxicity, but what if there is nitrate toxicity in the same plants?
4- Line 59, There are three primary sources of nitrogen nitrate (NO3 - ), ammonium (NH4 + ), and urea. Although nitrate and ammonia are predominant nitrogen forms in the soil, how are you classifying urea as a major source of nitrogen, unless used as a fertilizer?
5- In the introduction section, add a short description of Ssamchoo plants and their relevance to the current study.
6- In some of the figures, the statistical analysis is missing (Figure 4 a,b; Figure 9 a,c). Moreover, in some figures, I have a doubt about the statistical test used (Figure 8), kindly check it again. You can use Tukey’s test in place of DMRT as it is more reliable.
Reviewer 2 Report
1. The title of the article is that adding silicon can alleviate ammonium toxicity, but the abstract focuses on the first experiment, so it needs to be revised to add a description of the role of silicon in this experiment.
2. Part 2 is the result, the following label is wrong.
3. What is the nutritional content of Peters? Why do commercial nutrient solutions cause plant toxicity?
4. Is the nitrogen concentration of 13mmol L-1 too high in GNU1, which would cause excess nitrogen fertilizer?
5. What standard is the added silicon concentration (10.7m) based on?
6. The analysis results in Figure 11 are incorrect.
Reviewer 3 Report
According to the manuscript, Alleviation of ammonium toxicity and suppressed growth in Ssamchoo grown in home cultivation system by silicon supplementation to nutrient solution. The experimental hypothesis was to study the effect of silicon on the reduction of ammonia toxicity in Ssamchoo. However, studies have not shown any clear results. Please find the reviewer’s comments below.
1. The abstract is too long. The author should summarize the main idea of the work done as well as the results or benefits of the research so that the reader can easily understand.
2. The authors should explain the reasoning behind each experiment and describe the key reasons discovered in previous trials for subsequent trials. Because the results of the experiments are not continuous. The authors should state the reasons for each experiment and should describe the results in more detail.
3. The experimental method should be written for the reader to understand easily, especially the plant nutrients used should show the information in the table for easy understanding.
4. Most of the experimental results were to find suitable nutrients for plant growth. The results are unclear to conclude that silicon can reduce ammonia toxicity in plants. As in experiment 3, it was clearly seen that the growth effects of different nutrient plants with the addition of silicon were not significantly different.
5. The authors should clarify the findings from this study and what is different about the new findings from other studies. This is because there are many reports on the effect of silicon in reducing ammonia toxicity in plants.
Round 2
Reviewer 2 Report
Accept in present form.
Author Response
Thank you for your time spent reviewing the revised version. We accept your suggestion and the paper is much better now.
Reviewer 3 Report
The content has been edited and improved by the author. However, the majority of the experimental results were aimed at identifying suitable nutrients for Ssamchoo growth. As the authors responded, 50:25:25 treatment showed a significant effect of silicon. The authors should conduct more experiments to determine the effect of silicon on ammonia toxicity. Because the silicon effect experiments were too small and the results were only shown by looking at the morphology of the plants, this does not prove that silicon reduces ammonia toxicity. The authors should be able to demonstrate the key mechanism of silicon in reducing ammonia toxicity.
Author Response
Thank you for your time and comment on the revised version. We also agree with your suggestion, but we will leave the portion of elucidating mechanism to future experiments.